# Myc-Related Mitochondrial Activity as a Novel Target for Multiple Myeloma

**DOI:** 10.3390/cancers13071662

**Published:** 2021-04-01

**Authors:** Alejandra Ortiz-Ruiz, Yanira Ruiz-Heredia, María Luz Morales, Pedro Aguilar-Garrido, Almudena García-Ortiz, Antonio Valeri, Carmen Bárcena, Rosa María García-Martin, Vanesa Garrido, Laura Moreno, Alicia Gimenez, Miguel Ángel Navarro-Aguadero, María Velasco-Estevez, Eva Lospitao, María Teresa Cedena, Santiago Barrio, Joaquín Martínez-López, María Linares, Miguel Gallardo

**Affiliations:** 1H12O-CNIO Hematological Malignancies Clinical Research Unit, CNIO, 28029 Madrid, Spain; maortiz@cnio.es (A.O.-R.); yanira_heredia@altumsequencing.com (Y.R.-H.); mlmorales@cnio.es (M.L.M.); paguilar@cnio.es (P.A.-G.); a.garcia.imas12@h12o.es (A.G.-O.); avaleri27@yahoo.es (A.V.); gvaneg@hotmail.com (V.G.); lmorenos@salud.madrid.org (L.M.); gimenezsa.imas12@h12o.es (A.G.); manavarro@cnio.es (M.Á.N.-A.); mvelasco@tcd.ie (M.V.-E.); santiago_barrio@altumsequencing.com (S.B.); jmarti01@med.ucm.es (J.M.-L.); mlinares@ucm.es (M.L.); 2Hematology Department, Hospital Universitario 12 de Octubre, 28041 Madrid, Spain; mariateresa.cedena@salud.madrid.org; 3Pathology Department, Hospital Universitario 12 de Octubre, 28041 Madrid, Spain; carmen.barcena@salud.madrid.org (C.B.); rgarciamartin4@salud.madrid.org (R.M.G.-M.); 4CNIO–Lilly Cell Signalling and Immunometabolism Section, CNIO, 28029 Madrid, Spain; eplospitao@cnio.es; 5Biochemistry and Molecular Biology Department, Pharmacy School, Universidad Complutense de Madrid, 28040 Madrid, Spain

**Keywords:** multiple myeloma, mitochondria, tigecycline, MYC

## Abstract

**Simple Summary:**

Multiple myeloma represents the cancer with the 21st highest global incidence. The rapid acquisition of drug resistance to proteasome inhibitors requires a deep knowledge on the mechanisms involved in proliferation in order to provide novel targets for the disease. The aim of our study was to characterize the mitochondrial activity in primary multiple myeloma (MM) cells along the course of the disease and to provide a therapeutic alternative to inhibit Myc function by blocking OXPHOS metabolism. We confirmed MM patients show enhanced mitochondrial activity linked to c-Myc expression. The use of tigecycline provides evidence for a novel strategy addressing c-Myc functionality by mitochondrial activity inhibition.

**Abstract:**

Mitochondria are involved in the development and acquisition of a malignant phenotype in hematological cancers. Recently, their role in the pathogenesis of multiple myeloma (MM) has been suggested to be therapeutically explored. MYC is a master regulator of b-cell malignancies such as multiple myeloma, and its activation is known to deregulate mitochondrial function. We investigated the impact of mitochondrial activity on the distinct entities of the disease and tested the efficacy of the mitochondrial inhibitor, tigecycline, to overcome MM proliferation. COXII expression, COX activity, mitochondrial mass, and mitochondrial membrane potential demonstrated a progressive increase of mitochondrial features as the disease progresses. In vitro and in vivo therapeutic targeting using the mitochondrial inhibitor tigecycline showed promising efficacy and cytotoxicity in monotherapy and combination with the MM frontline treatment bortezomib. Overall, our findings demonstrate how mitochondrial activity emerges in MM transformation and disease progression and the efficacy of therapies targeting these novel vulnerabilities.

## 1. Introduction

Multiple myeloma (MM) is a hematologic malignancy of plasma cells that accounts for 1% of all cancers and ~10% of all blood cancers [1]. Despite extensive efforts to improve the survival of patients with MM, current therapies are unsatisfactory and ~50% of patients quickly develop resistance and relapse [2]. Several novel treatments have been tested for relapsed MM, however, a better understanding of the molecular mechanisms that control the onset and progression of malignant MM is needed to develop more specific and effective interventions.

One of the most recognized molecules involved in the pathogenesis of MM is MYC gene deregulation. MYC (v-myc myelocytomatosis viral oncogene homolog, c-MYC) is an oncogene with many important functions at the cellular and nuclear levels, whose activation regulates several genes involved in mitochondrial transcription, mitochondrial translation, protein import, and complex assembly [3,4]. MYC has been described as a proto-oncogene involved in many cancers, including leukemia and lymphoma [5,6,7]. In MM, it is known that aberrant expression of MYC is due to recurrent chromosomal alterations [8] and previous studies have demonstrated that its expression is involved in MM progression from early stages of plasma cell dyscrasias, including monoclonal gammopathy of unknown significance (MGUS) and smoldering multiple myeloma (SMM), to MM [9]. Several authors have reported that MYC deregulation is involved in disease progression and has been related with poor progression-free survival in MM patients [10]. Moreover, MYC direct transcription has been described as promoting changes in mitochondrial dynamics that influence therapy resistances [11]. However, MYC is a protein that traditionally could not be targeted pharmacologically. In this context it has been proposed that synthetic lethal interactions with MYC could be exploited as an alternative to target it. In this case, Hsieh has proposed that nutrient deprivation leads to cell death as MYC-overexpressing cells are glucose- and glutamine-addicted [12]. Previous studies have reported that MM is dependent on glutamine for survival, and that also depends on MYC expression [13]. It is known that MYC overexpression has exaggerated effects on mitochondrial biogenesis, oxidative metabolism, and sensitivity to OXPHOS inhibitors [14]. Previous studies have demonstrated that MM patients have increased mitochondrial biogenesis compared to normal plasma cells, and corroborated that MYC promoted this process [15]. In this context, control of MYC expression has been proposed to be a meaningful achievement in the fight against MM [9]. As strategies to directly target MYC have not been reached until now, key targets involved in MYC deregulation have been exploited as new approaches to treat MYC-driven malignancies.

Mitochondria are bioenergetics and biosynthetic organelles that control several crucial cellular pathways, including growth, proliferation, and apoptosis [16,17]. Numerous studies have evaluated the role of mitochondrial DNA (mtDNA) alterations in cancer, and modifications in mitochondrial content, structure, function, activity [5,18], and respiration [19], all of which seem important for cancer progression. Indeed, mitochondria are essential for tumor formation and growth, and their dysfunction promotes tumor cell metastasis [20]. Oxidative phosphorylation (OXPHOS) was traditionally believed to be downregulated in all cancers, but recent studies have demonstrated that some cancers show an upregulation, particularly in hematological cancers, reviewed in [21,22]. Accordingly, cancer has been characterized as an altered state of energy metabolism involving genetic alterations in nuclear DNA (nDNA), in mitochondrial DNA (mtDNA), and in mtDNA copy number (mtDNA CN) [20,23,24], and metabolic reprogramming based on shifts in OXPHOS activity, representing a hallmark of malignancy and cancer progression [23].

There is a crucial need to identify new cellular targets and/or pathways that regulate metabolic reprogramming during MM cell growth and progression and, in this respect, addressing mitochondrial activity represents a novel pathway to target this vulnerability [24].

Tigecycline is a glycylcycline class antibiotic with similar structure to tetracyclines. The mechanism of action of tigecycline is the protein translation inhibition by 30S ribosomal unit binding, with high activity against gram-negative and gram-positive bacteria. However, tigecycline is also known to inhibit mitochondrial oxidative phosphorylation, by suppressing COXI and COXII, and inhibiting mitochondrial activity, translation, and biogenesis. This, and other molecular mechanisms that are not well-known, impacts against cancer cells, and its use in different tumors has been suggested [25].

Here, we explored the enhanced mitochondrial activity features in MM development and tested the therapeutic effect of the OXHPOS inhibitor, tigecycline, on MM, a MYC-driven malignancy. Our study identified a novel vulnerability of MYC through mitochondrial activity regulation to avoid MM progression.

## 2. Materials and Methods

### 2.1. Myeloma Cell Lines and Primary Multiple Myeloma Samples

A total of 110 samples from 83 patients were collected in accordance with the Declaration of Helsinki and with the approval of the Ethics Committee of the Hospital 12 de Octubre (Madrid, Spain). Written consents were obtained from patients for tissue collection. Clinical data of patients are summarized in Table 1 and Appendix A. All primary MM specimens were obtained from bone marrow, and cylinders and aspirates were processed for further analysis. Cylinders were fixed and processed for COXII (cytochrome c oxidase subunit II) immunohistochemistry. Aspirates were processed by standard Ficoll-Paque density centrifugation. Plasma cells were purified by magnetic-activated cell sorting with CD138+ microbeads (Miltenyi Biotec, Auburn, CA, USA). Fresh bone marrow myeloma cells (BMMCs) were cultured in RPMI medium (Lonza, Basel, Switzerland) supplemented with 10% fetal bovine serum (FBS) (Hyclone, ThermoFisher Scientific, Waltham, MA, USA), penicillin/streptomycin (Lonza, Basel, Switzerland), and 6% plasma from the same patient.

Three human myeloma cell lines (JJN3 S, L363, and OPM2) were obtained from the DSMZ cell repository (Braunschweig, Germany). Cell lines were expanded and cryogenically frozen upon acquisition to establish stocks in liquid nitrogen until use. The RPMI-8226 cell line was kindly provided by Christopher Driessen (Kantonsspital, St. Gallen, Switzerland) and NCI-H929 cells were a kind gift from Joaquin Teixidó (CIB, Madrid, Spain). JJN3 BR cells were established by dose escalation of bortezomib once-weekly for a period of five months. JJN3 S cells were lentivirally transduced with a third-generation pRLL-luc/GFP virus and helper plasmids (pMDLg/RRE, pREV, and VSV-G). Vector titration was performed using dilutions of lentiviral supernatants. Cells expressed firefly luciferase and GFP in a fusion protein driven by the CMV promoter. GFP+ cells were detected by flow cytometry and quantitative real-time PCR (qPCR), followed by sorting on a FACS BD INFLUX cell sorter (BD Biosciences, Franklin Lakes, NJ, USA). All cells were maintained at 37 °C in a humidified incubator, in an atmosphere of 5% CO_2_, and passaged every 2–3 days. Cells were cultured in RPMI medium supplemented with 10% fetal bovine serum (FBS) and penicillin/streptomycin.

### 2.2. MYC Gene Expression Analysis

We analyzed RNAseq data from a total of 770 samples of patients with MM at diagnosis from the CoMMpass repository. The data were generated as part of the publicly available Multiple Myeloma Research Foundation Personalized Medicine Initiatives (https://research.themmrf.org, accessed on 22 April 2020) with the identifier IA14, under “Code availability”. We ran the molecular gene view on the entire dataset to study the expression of MYC. Kaplan–Meier curves of PFS were plotted with balanced stratification of high and low expression of the gene studied.

### 2.3. RNA and DNA Extraction and Quantitative PCR (qPCR)

We extracted RNA and DNA from purified plasma cells (CD138+) using the AllPrep DNA/RNA Mini Kit (Qiagen, Valencia, CA, USA). Evaluation of the mtDNA CN was performed with DNA extracted from 5 MM cell lines (OPM2, L363, RPMI8226, NCI-H929, and JJN3) and the ratio between mtDNA and nDNA was examined using the 12 S mitochondrial probe and the RNAsaP nuclear probe, respectively, as described [26]. Gene expression was studied from the extracted RNA and processed into cDNA to quantify the expression of MYC (ThermoFisher Scientific, Waltham, MA, USA), while beta-D-glucuronidase (GUSB, ThermoFisher Scientific, Waltham, MA, USA) was used as a housekeeping gene. This dataset included 47 samples from Hospital 12 de Octubre patients: 9 MGUS, 9 SMM, 18 NDMM, 11 RRMM, and 3 MM cell lines, which were profiled using TaqMan probes. qPCR was performed on an ABI PRISM 7900HT instrument (Applied Biosystems; Grand Island, NY, USA) with TaqMan gene expression (Life Technologies, Grand Island, NY, USA) assays in triplicate using SDS 2.2 software (Applied Biosystems, Framingham, MA, USA). The relative amount of targets was determined by the comparative Ct method [27].

### 2.4. Optical Microscopy

For immunochemistry analysis, paraffin-embedded sections (0.5 µm) were deparaffinized with xylene, rehydrated with graded ethanol, and stained with an antimitochondria antibody (MTC02; Abcam, Cambridge, UK), followed by DAB development. Sections were then counterstained with hematoxylin and mounted with DPX (Sigma-Aldrich, Madrid, Spain). Images were classified on an intensity scale of 0–2, with 2 being the highest intensity stain (2: positive cells [Grade II]; 1: positive cells [Grade I]; 0: negative cells). Stained whole samples were assessed under double-blinded conditions. All slides were evaluated in conventional light-field microscopy, objective 40X, using an optical microscope (Olympus AX70), and analyzed with the Olympus Cellsens software. The dataset included 4 MGUS, 7 SMM, 13 NDMM, and 11 RRMM samples. For the histoenzymatic assay, fresh samples (CD138 + BMMCs) were spun onto a slide by cytospin centrifugation (700× *g*, 5 min, Shandon Single Cyto Slides™; ThermoFisher, Waltham, MA, USA). Thereafter, slides were incubated in a rehydrated COX (cytochrome C oxidase enzyme) solution (Bio-Optica, Milan, Italy) at 37 °C for 1 h and 30 min, and were then counterstained with Carazzi’s hematoxylin for 2 min (Panreac, Barcelona, Spain). Detection was performed using the same microscope Olympus AX70 objective 100X on 18 patient samples (4 MGUS, 5 SMM, 5 NDMM, and 4 RRMM) and 5 MM cell lines (OPM2, L363, RPMI8226, NCI-H929, and JJN3). Quantification of the stained cells was manually assessed under double-blinded conditions, providing for each slide a score based on a scale from 0 to 4 (Grade 0: no stain; Grade 1: low stain; Grade 2: medium stain; Grade 3: high stain; Grade 4: very high stain).

### 2.5. Multiparametric Flow Cytometry Analysis

Erythrocyte-lysed whole BMMCs, JJN3 S, and JJN3 BR cells lines were studied by flow cytometry. All events from the distinct conditions applied to BMMCs were analyzed and 10,000 events for each condition in the case of experiments with cell lines. Plasma cell (PC) population CD38+/CD138+ was assessed to detect variations of GeoMean PCs after 48-h treatment with tigecycline (3.7 µM), bortezomib (2 nM), and their combination. Cells were incubated for 30 min at room temperature with CD38 FITC (BD Biosciences, Franklin Lakes, NJ, USA), and CD138 PECy7 (Biolegend, San Diego, CA, USA). Data from the different treatments were normalized to samples with no treatment, Moreover, mitochondrial content and membrane potential were analyzed with MitoTracker Green and Mitotracker Red probes from ThermoFisher Scientific (Waltham, MA, USA), respectively. JJN3 S, JJN3 BR cells, and BMMCs were incubated for 10 min at 37 °C with 150 nM MitoTracker Green and with 150 nM MitoTracker Red, followed by a PBS wash. Additionally, Mitotracker Green and Mitotracker Red were assessed over JJN3 S and JJN3 BR after 48-h exposure to tigecycline (3.7 µM), bortezomib (2 nM), and their combination or JQ1 (78 nM), bortezomib (2 nM), and their combination. Death cells were stained with DAPI.

### 2.6. In Vitro and Ex Vivo Drug Assays

The effect of tigecycline (Selleckchem, Houston, TX, USA) on cell viability and mitochondrial activity was assessed in MM cell lines (JJN3, JJN3 BR). Bortezomib was also purchased from Selleckchem and both drugs were diluted in DMSO. JQ1 was kindly provided by the Pharmacy Department at Hospital 12 de Octubre. Cells were seeded at 1 × 10^6^ cells/mL in 96-well plates. Cell lines and CD138+ BMPCs were treated for 48 h with bortezomib and/or tigecycline (0.1% DMSO), JQ1, or vehicle. For drug synergy calculation, a range of serial dilutions of tigecycline were made across the IC50 value. For bortezomib, only one concentration (2 nM) was selected for the combinatory assays. The agents were then added simultaneously for 48 h and cell viability assays were performed. Additionally, we performed a “conditioning + treatment” assay, with a 48-h pretreatment of tigecycline (33, 11, and 3.7 µM) followed by 48-h bortezomib treatment of five serial dilutions. Quantitative analysis of dose–effect relationships of bortezomib and tigecycline were determined after measurement of cell growth using the cell-permeant dye Calcein-AM (Sigma-Aldrich, Madrid, Spain) (100 µM). The signal intensity of the stained cells was monitored at 496 nm (excitation) and 516 nm (emission) using the EnSpire Multimode Plate Reader (Perkin Elmer, Waltham, MA, USA) [28]. Viability data were normalized to those of DMSO controls. Potential synergistic or additive effects were calculated using CompuSyn software (ComboSyn Inc., Biosoft; Cambridge, UK). Drug synergism, addition, and antagonism effects were defined by combination index values of <1.0, 1.0, and >1.0, respectively. The Chou–Talalay Combination Index Theorem was used to calculate the combination index (CI) of bortezomib and tigecycline [29].

### 2.7. Western Blotting

Cells were harvested after stimulation and then lysate in RIPA buffer. Lysates were cleared by centrifugation (14,000× *g*, 4 °C, 30 min) and protein concentration was determined by Bradford Assay (Bio-Rad Laboratories, Hercules, CA, USA). Samples were normalized to 46.7 µg of total proteins and loaded onto 10% SDS-PAGE gels followed by transfer onto a PVDF membrane. Blots were incubated with primary and secondary antibodies (listed in Appendix A). An antibody to β-actin or GAPDH was used to check for equivalent protein loading. Membranes were developed by enhanced chemiluminescence (Clarity Western ECL substrate; Bio-Rad Laboratories, Hercules, CA, USA) and the protein bands were detected with a ChemiDoc MP instrument (Bio-Rad, Hercules, CA, USA). Densitometry was performed with Image Lab software, normalized to the signal of β-actin or GAPDH, and normalized to the control, set as one. Uncropped Western blot data are attached as Appendix A.

### 2.8. Animal Model

A MM xenograft mouse model was used for the in vivo study. The animal study was conducted according to the guidelines of the Institutional Animal Care and Use Committees of the Comunidad de Madrid, Spain, under an approved protocol (PROEX 023/17). Female 6–8 week-old NOD.Cg-Prkdcscid Il2rgtm1Wjl/SzJ (NSG) mice (Vivotecnia, Madrid, Spain) were inoculated intravenously by tail vein injection with JJN3-ffLuc-GFP cells (1 × 10^6^). Four days later, mice were randomized into four groups of 10 animals—(1) the first group received intraperitoneal injections of vehicle (0.9% saline solution) five days a week; (2) the second group received 0.2 mg/kg bortezomib intraperitoneally five times weekly for four weeks; (3) the third group received intraperitoneal injections of tigecycline five times weekly for four weeks at increasing doses, 50–50–75–100 mg/kg/week (starting at 50 mg/kg for the first two weeks, followed by 75 mg/kg for the third week, and 100 mg/kg for the fourth week); (4) the fourth group was treated with a combination of bortezomib and tigecycline, five times weekly, the first two weeks with 0.2 mg/kg bortezomib and 50 mg/kg tigecycline, and the following weeks with increasing doses of tigecycline (third week: 75 mg/kg; fourth week: 100 mg/kg). Five mice of each treatment group were monitored for tumor burden, distribution, and engraftment every week by IVIS whole-body bioluminescence imaging. Bioimaging of MM burden in vivo was performed by IVIS Imaging system (Caliper Life Science, Hopkinton, MA, USA). Before the acquisition, Luciferin (D-luciferin monosodium salt, ThermoFisher, Waltham, MA, USA) was prepared in PBS and 150 mg/kg and was injected intraperitoneally, with luminescence signal detection 10 min later. The signal data were analyzed using Living Image 4.2 (Caliper Life Science, Hopkinton, MA, USA) software.

### 2.9. Statistical Analysis

Statistical analysis was performed using GraphPad Software Inc. (version 6.0, La Jolla, CA, USA). All data are presented as the mean ± SEM and are representative of three independent experimental replicates (*N* = 3). Normally distributed data were analyzed by unpaired Student’s *t*-test or one-way ANOVA. For nonparametric data, the Mann–Whitney test was used to compare two groups. For more than two independent samples, statistical comparisons were carried out using the nonparametric method Kruskal–Wallis test, when the distribution was not normal. The major clinicopathological characteristics and available treatment information of the patients are presented in Table 1 and Appendix A and summarized by frequency and percentage. PFS were analyzed by Kaplan–Meier curves. *p*-values < 0.05 were considered to be statistically significant.

## 3. Results

### 3.1. Patients with Newly Diagnosed and Relapsed MM Exhibit Elevated Myc Expression Features

We first assessed MYC gene expression in a dataset of patients with NDMM. Using RNAseq CoMMpass data of 770 patients [30], we observed that PFS was significantly decreased when MYC was overexpressed (hazard ratio [HR] 1.18), as illustrated in Figure 1A. Additionally, we performed differential MYC expression along the distinct MM entities. The pattern of increased overexpression of MYC was progressive, revealing a significant increase in those patients with NDMM, and a more noticeable one in RRMM, as is evident in Figure 1B.

As the activity of respiratory enzymes is a well-defined approach for investigating mitochondrial dysfunction [31], we next quantified the expression of the cytochrome c oxidase subunit II (COXII) and the activity of the enzyme COX, which is essential for mitochondrial function. COXII is one of the three mtDNA-encoded subunits of the respiratory complex IV and part of the functional core of the enzyme complex. Quantification of the protein by immunohistochemistry (IHC) in Figure 1C revealed the overexpression of COXII in progressed MM groups (*p* value < 0.0001). We also confirmed in Figure 1D that COX enzyme activity was significantly higher in the NDMM and RRMM groups than in the MGUS and SMM groups (*p* value < 0.0001), as revealed by the greater intensity of the stain. As previously reported, there is a clear correlation between COX activity and the protein content of COXII [32]. Given its role as the rate-limiting step of the respiratory chain [33], the overexpression of COXII is an indicator of the high oxidative capacity of the cells.

Additionally, we assessed total mitochondrial mass and membrane potential in CD138 + BMMCs using MitoTracker Green and Red, respectively. Both probes revealed higher staining in patients with NDMM than in patients with a premalignant stage (MGUS), as confirmed in Figure 1E. See Table 1, Table 2 and Appendix A, for patient clinic characteristics pertinent to the entire analysis. Overall, these results point to a role for mitochondria in MM, as revealed by an increase in the mitochondrial burden that enhances mitochondrial activity as the disease progresses.

### 3.2. Mitochondrial Inhibition Is Cytotoxic to MM Cells In Vitro and Synergizes with Bortezomib

To determine the most appropriate in vitro model to test the effect of mitochondrial inhibition on MM, we first characterized five MM cell lines according to their mtDNA load and mitochondrial COX activity, as shown in Figure 2A,B. For viability assays, we selected the JJN3 cell line, which displayed the highest mitochondrial burden and activity and also high levels of MYC expression (Figure 2C).

We next interrogated high mitochondrial activity as a novel vulnerability in c-Myc-driven MM. First, we tested the impact of the mitochondrial inhibitor tigecycline alone or in combination with the frontline drug bortezomib in JJN3 cells, which are bortezomib-sensitive (JJN3 S). Inhibitory assays represented in Figure 3A demonstrated cytotoxic effects of tigecycline in monotherapy (IC50 = 21.3 µM) and in combination with bortezomib, at 48 h. To quantify the effect of the combination, we performed a synergy analysis, applying the Chou–Talalay method. All doses studied showed high combination indexes (CI < 0.5). Additionally, we tested the sensitizing effect of tigecycline to bortezomib. A 48-h treatment with tigecycline before a consecutive 48-h treatment with bortezomib highlighted the possibility to also target progressed patients, due to successful sensitization to bortezomib with prior tigecycline treatment as shown in Figure 3B.

To identify the mechanism of action of mitochondrial inhibition by tigecycline, we determined the protein levels of MYC and complexes of the respiratory chain. Western blot (WB) analysis, represented in Figure 3C, revealed a remarkable decrease in the levels of complex I and IV subunits upon exposure to tigecycline. Hence, the combination with bortezomib showed a similar reduction of the complexes’ expression and additional reduction of c-Myc expression.

To corroborate the finding that a reduction in complex IV abundance was concomitant with a reduction in enzyme activity, we performed a histoenzymatic assay of cytochrome c oxidase (HE-COX). Interestingly, we observed that tigecycline alone or in combination with bortezomib led to a loss of COX activity, as shown in Figure 3D.

Then, we investigated mitochondrial mass and membrane potential, two parameters of mitochondrial status. Mean fluorescence intensity showed a decrease in both parameters in those cells treated with tigecycline and the combination, as represented in Figure 3E. By contrast, the fluorescence was essentially unchanged in the bortezomib-treated (Figure 3E) and carfilzomib-treated (Appendix A) groups. Overall, these data indicate that bortezomib reduces c-Myc, while tigecycline reduces mitochondrial activity, and the combination triggers both c-Myc and mitochondrial activity reduction.

Additionally, we assessed the mitochondrial inhibitory effect of tigecycline on the resistant JJN3 cell line (JJN3 BR). We found a cytotoxic effect using similar doses as those employed in the sensible counterpart (Appendix A). Moreover, the assessment of mitochondrial features of JJN3 BR cells exposed to tigecycline or the combination with bortezomib revealed also the same trends found with JJN3, as the complexes of the respiratory chain, the cox activity, and the mitochondrial membrane potential appeared reduced, as shown in Appendix A.

To confirm that mitochondrial activity blockade is an alternative to inhibit a c-Myc-driven mechanism in an indirect manner, and thereby a novel vulnerability, we examined the effects of an alternative c-Myc indirect inhibition independent of mitochondria by the use of JQ1, a bromodomain BRD4 inhibitor that reduces c-Myc levels. The efficacy of JQ1 in monotherapy in JJN3 S and JJN3 BR cells was confirmed by a similar decrease of viability (Appendix A). Moreover, the cytotoxicity of JQ1 was enhanced when combined with bortezomib (Appendix A). WB confirmed a decrease in c-Myc levels in cells treated with bortezomib, JQ1, and bortezomib + JQ1. Complex I and IV of OXPHOS expression were not reduced in JQ1-treated cells, nor in the combination with the bortezomib group. Moreover, HE-COX and MitoTracker Red staining demonstrated no changes in mitochondrial activity, suggesting that mitochondria regulation is a vulnerability for MYC, however, short-term direct inhibition of MYC does not control the reprogrammed mitochondrial activity (Appendix A).

### 3.3. Tigecycline Is Active against Primary Myeloma Cells in Vitro and in a Murine MM Model

We tested the efficacy of tigecycline alone or in combination with bortezomib on primary BMMCs from patients. Details of the patient characteristics, sample analysis performed, and patient therapy regimens are summarized in Table 1, Table 2 and Appendix A, respectively. We found that CD38+/CD138+ plasma cells from patients with NDMM had reduced viability with tigecycline and its combination with bortezomib compared to no treatment condition. Reduction of viability by tigecycline correlated with a decrease in mitochondrial activity analyzed by MitoTracker Red staining, as shown in Figure 4A,B. Overall, these results show that MM primary cells are sensitive to tigecycline, which kills the malignant cells by ablating a molecular mechanism of c-Myc oncogenesis, the mitochondrial activity pathway that is implicated on MM progression.

Given these in vitro findings, we investigated whether tigecycline treatment alone or in combination with bortezomib could be effective in an in vivo animal model representing high mitochondrial activity. To determine the therapeutic effect of tigecycline in vivo, we assessed its antimyeloma efficacy in a murine xenograft model, as previously described [34]. Xenografted mice with the bioluminescent cell line JJN3-ffLuc-GFP, which recapitulates the aggressiveness of the disease and the elevated mitochondrial activity, were treated starting on day four after intravenous engraftment. We found that tigecycline treatment significantly decreased the burden of the disease measured by serial, noninvasive bioluminescent imaging, as shown in Figure 5A. In agreement with this, we found that the survival of animals treated with tigecycline (*p* = 0.0003) and the combination of tigecycline and bortezomib (*p* = 0.0048) was significantly greater than survival obtained in the control group (Figure 5B). These results demonstrate that tigecycline, alone or in combination with low doses of bortezomib, enhances the survival of MM-engrafted mice, indicating that tigecycline may be valuable for MM patients presenting signs of high mitochondrial activity at diagnosis and to prevent the metabolic reprogramming, in part, triggered by proteasome inhibitor treatments that contribute to cell malignancy.

## 4. Discussion

Our data show that mitochondrial activity increases in primary MM cells as the disease progresses. Mitochondrial features allow us to characterize patient samples at different stages of the disease and appeared to be related to MYC-driven malignancy, suggesting a novel indirect target of MYC to overcome MM proliferation.

Malignant plasma cells exhibit an erratic behavior determined by mitochondrial metabolism, a well-characterized biological mechanism in the context of apoptosis [24]. The bone marrow microenvironment has been shown to sustain MM cells via intercellular mitochondrial transfer, which stimulates OXPHOS and cancer progression [35].

Similarly to our findings, the observation of MYC overexpression across the massive RNAseq analysis and the correlation with patient poor prognosis was previously reported with a decreased in the overall survival in a cohort of MM patients [9]. As MYC has the capacity to regulate mitochondrial biogenesis, we found that mitochondrial mass, and consequent mitochondrial membrane potential is augmented in patients showing disease progression. We corroborated these results with COX activity in MM primary cells (CD138 + BMMC) with a histoenzymatic assay commonly used for muscle biopsies [36].

To demonstrate the mitochondrial reprogramming influenced by MYC overexpression in progressed MM cells, and its potential as a therapeutic target, we used the antibiotic tigecycline, which inhibits protein synthesis in both bacterial and mitochondria-encoded proteins, due to the similarity of mitochondrial and bacterial ribosomes [37]. Previous studies have demonstrated the therapeutic efficacy of tigecycline against some cancers, such as B-cell malignancies [37], primary acute myeloid leukemia [38,39], and myeloma [40]. Using an in vitro model of progressed MM with high mtDNA, high MYC expression, high mitochondrial activity, and ex vivo primary MM cells (CD138 + BMMCs), we confirmed the cytotoxic effect of tigecycline. Moreover, additional results were reported from in vivo experiments, as tigecycline significantly delayed disease progression [40]. Additionally, the in vitro MM bortezomib resistance (BR) model demonstrated a similar sensitivity to tigecycline, as seen in the bortezomib-sensitive counterparts. To explore the therapeutic options for MM, we studied the tigecycline and bortezomib combination, finding synergism in the combination against bortezomib-sensitive cells with low doses of tigecycline (3.7 µM) and bortezomib (2 nM). Thus, tigecycline and other mitochondrial inhibitors could be useful in metabolically reprogrammed malignancies. For instance, the Bcl-2 inhibitor venetoclax has been shown to inhibit mitochondrial respiration [41], although it appears to have higher cytotoxicity in myeloma plasma cells with reduced mitochondrial respiration. The increased OXPHOS in MM would be a drawback for venetoclax treatment, but the addition of tigecycline to a venetoclax regimen could be a therapeutic opportunity in MM or other B-cell malignancies, especially with a MYC contribution [42]. Tigecycline elicited its effects on mitochondrial activity related to MYC overexpression, consistent with the findings of other authors [43].

MYC is a master proto-oncogene with high implication in myeloma, accordingly 20–50% of patients with MM show MYC translocations and 15–20% show amplifications of the gene [44,45]. MYC regulates a plethora of biological functions, including the stimulation of mitochondrial biogenesis by regulator genes, lately contributing to mitochondrial mass gain [46]. However, the same mitochondrial reprogramming driven by MYC renders novel vulnerabilities, as revealed by synthetic lethality [12]. We stated the efficacy of MYC abrogation in MM cells using JQ1 and tigecycline. Our data suggest that elevation in mitochondrial biogenesis and consequent mitochondrial activity in MM is a malignancy mechanism linked to MYC. Accordingly, the mitochondrial machinery could be a potential indirect target of “undruggable” MYC through effective alternative targets of common antibiotics [47] such as tigecycline [48] and clarithromycin, among others [49].

In the light of the anticancer potential of mitochondrial inhibitors targeting molecular mechanisms of MYC, tigecycline warrants further development, given its potentially low on-target toxicity, avoiding proteasome inhibitors resistances and/or MM relapse.

## 5. Conclusions

In summary, our results demonstrate that elevation of mitochondrial activity occurs in MM patients as the disease progresses, causing a noticeable enhancement in those patients that relapse. We provided a novel alternative for MYC inhibition by targeting mitochondrial activity, as an indirect mechanism to avoid MM proliferation, with the mitochondrial inhibitor tigecycline. Importantly, the tigecycline efficacy alone or in combination with bortezomib ex vivo and in vivo reduces c-Myc-driven MM vulnerabilities, which makes tigecycline attractive for MM therapy.

## Figures and Tables

**Figure 1 cancers-13-01662-f001:**
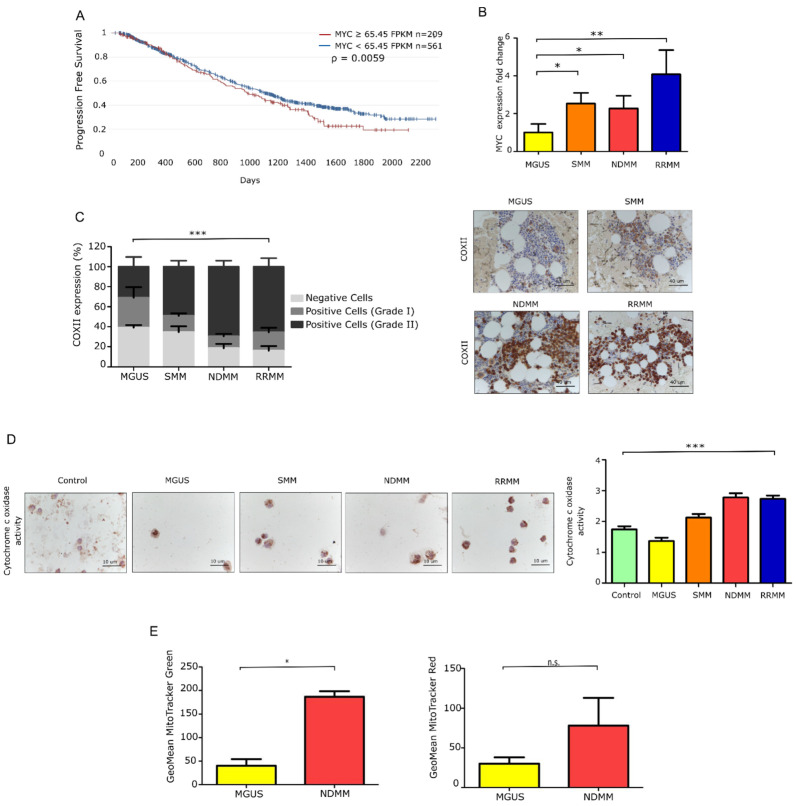
Mitochondrial features in primary MM cells from distinct myeloma entities. (**A**) Progression-free survival (PFS) of MM patients from the CoMMpass study (IA14) stratified according to MYC gene expression. (**B**) Fold-change in gene expression of MYC relative to GUS B expression (MGUS *N* = 9; SMM *N* = 9; NDMM *N* = 18; RRMM *N* = 11). Data are presented as mean values ± SEM of technical triplicates. (**C**) Intracellular expression of COXII in samples from MM patients (*N* = 4 MGUS, *N* = 7 SMM, *N* = 13 NDMM, and *N* = 11 RRMM). Microscopy analysis of the immunohistochemistry (IHC) slides from bone marrow samples (*p* value = 0.0001). Representative images addressing differences in COXII expression between patients at different stages of the disease. Scale bar, 40 μm. Data are presented as mean values ± SEM. Statistical significance was assessed using two-way ANOVA. (**D**) Histoenzymatic reaction to cytochrome c oxidase enzyme (HE-COX) in donors and patients across the different stages of the disease (*N* = 2 donor, *N* = 4 MGUS, *N* = 5 SMM, *N* = 5 NDMM and *N* = 4 RRMM). Relative HE-COX scores in fresh plasma cells (CD138+) ranging from Grade 0: no stain; Grade 1: low stain; Grade 2: medium stain; Grade 3: high stain; Grade 4: very high stain. Representative images show different intensity stains related to enzyme activity. Data are presented as mean values ± SEM. (**E**) Mitochondrial mass and membrane potential in MM patients (*N* = 3 MGUS, *N* = 3 NDMM), assessed by flow cytometry. Data represent the geometric mean ± SEM of three independent experiments. * *p* < 0.05, ** *p* < 0.001, *** *p* < 0.0001, n.s. no significance.

**Figure 2 cancers-13-01662-f002:**
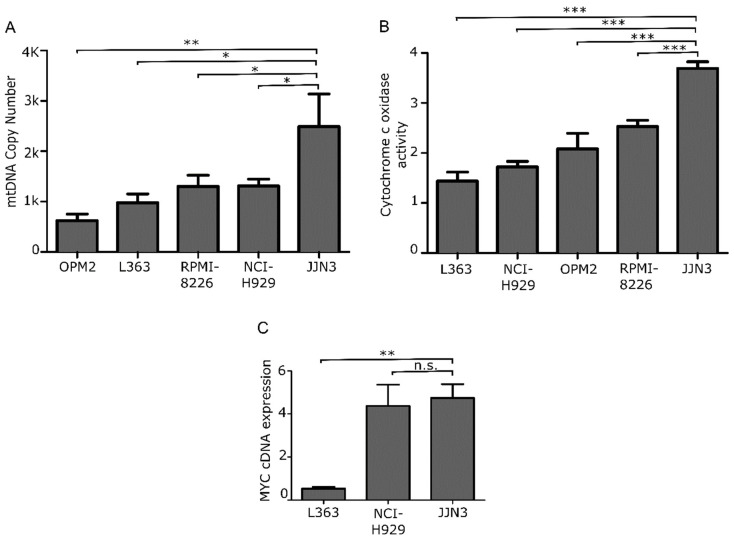
Mitochondrial features in MM cell lines. (**A**) mtDNA CN analysis and (**B**) HE-cox assay quantification in 5 MM cell lines. (**C**) Fold-change gene expression level of MYC in L363, NCI, and JJN3 cell lines relative to GUS B expression. Data are presented as mean values ± SEM of technical triplicates. * *p* < 0.05, ** *p* < 0.001, *** *p* < 0.0001, n.s. no significance.

**Figure 3 cancers-13-01662-f003:**
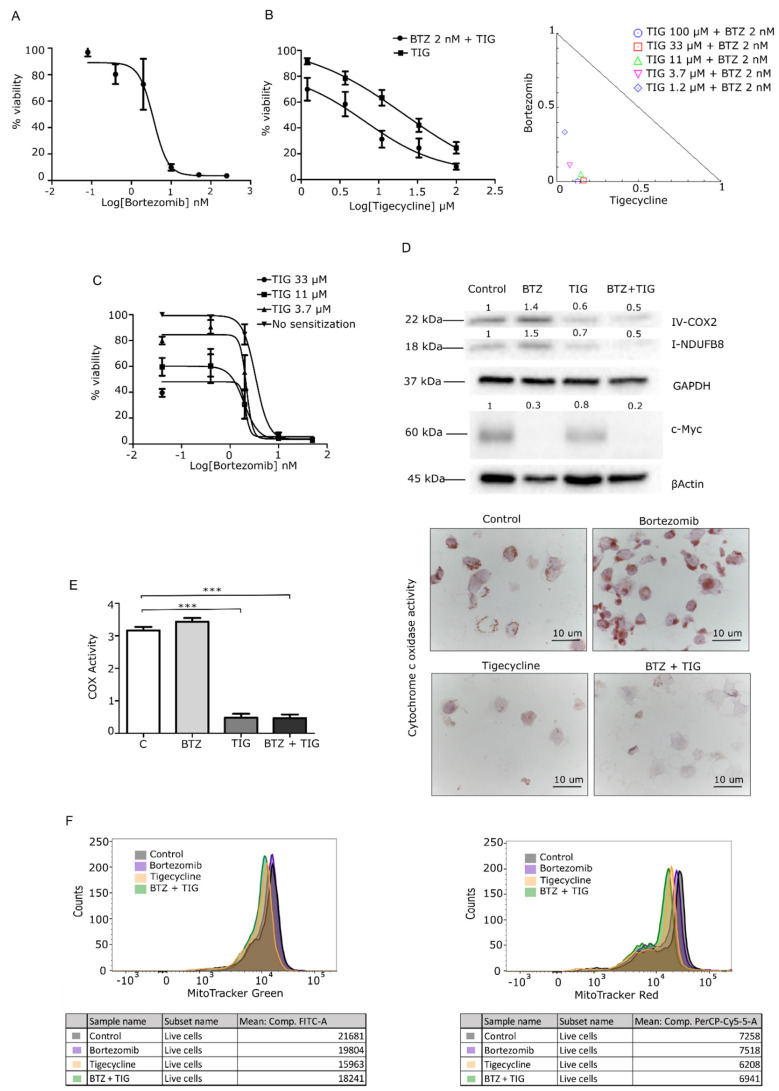
Tigecycline and the combination with bortezomib inhibits cancer cell growth and mitochondrial activity in the JJN3 cell line. (**A**) Cells were treated with a dose range of 0–250 nM of BTZ, and viability was determined using the calcein cytotoxic assay. (**B**) Cells were treated with a dose range of 0–100 μM of TIG with/without 2 nM BTZ, and viability was determined using the calcein cytotoxicity assay. TIG showed an IC50 of 21.3 µM in JJN3 cells, and the combination decreased threefold the IC50 value (6.6 µM). Isobologram plot of the combination; dots indicate the combinatory index (CI) of each combinatory dose. (**C**) IC50 curve after treating cells for 48 h with TIG for consecutive 48 h with bortezomib. Data are normalized to the negative control (DMSO) and presented as mean values ± SEM of technical triplicates. (**D**–**F**) molecular validation of mitochondrial inhibition of JJN3 cells treated with 3.7 µM TIG, 2 nM BTZ, or both (combination). All of the drug treatments were for 48 h; *N* = 1 independent biological repeat. (**D**) WB of respiratory chain complexes I and IV and c-Myc. (**E**) HE-COX analysis in >10 cells per sample. (**F**) Multiparametric flow cytometry (MFC) analysis of MitoTracker green (**left**, mitochondrial mass) and Red (**right**, mitochondrial membrane potential). Representative histograms show cell counts for each probe on the different treatment conditions. *** *p* < 0.0001.

**Figure 4 cancers-13-01662-f004:**
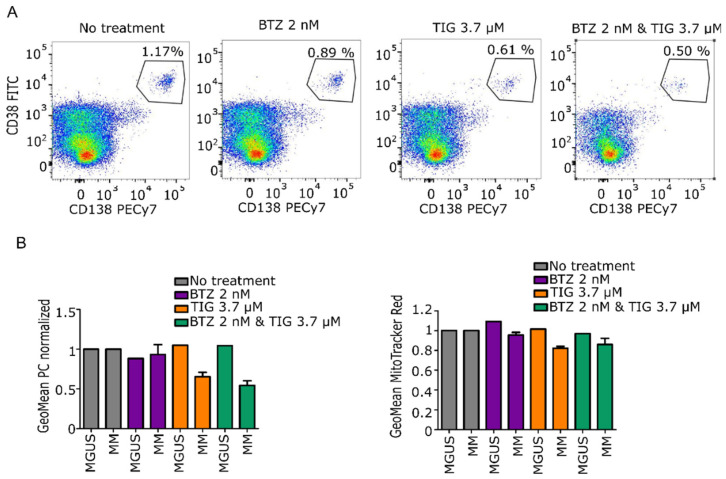
Tigecycline and the combination with bortezomib inhibits primary MM cell proliferation compared to premalignant entity MGUS. (**A**) Representative dot plots showing changes in the plasma cell population upon exposure to treatments. Numbers represent the proportion of CD38+ and CD138+ cells population. (**B**) **Left**: Bar chart reporting the mean of positive CD38+/CD138+ cells ± SEM population normalized against no treatment. **Right**: Bar chart reporting mean of MitoTracker Red expression ± SEM normalized against no treatment (MGUS *N* = 1; NDMM *N* = 3).

**Figure 5 cancers-13-01662-f005:**
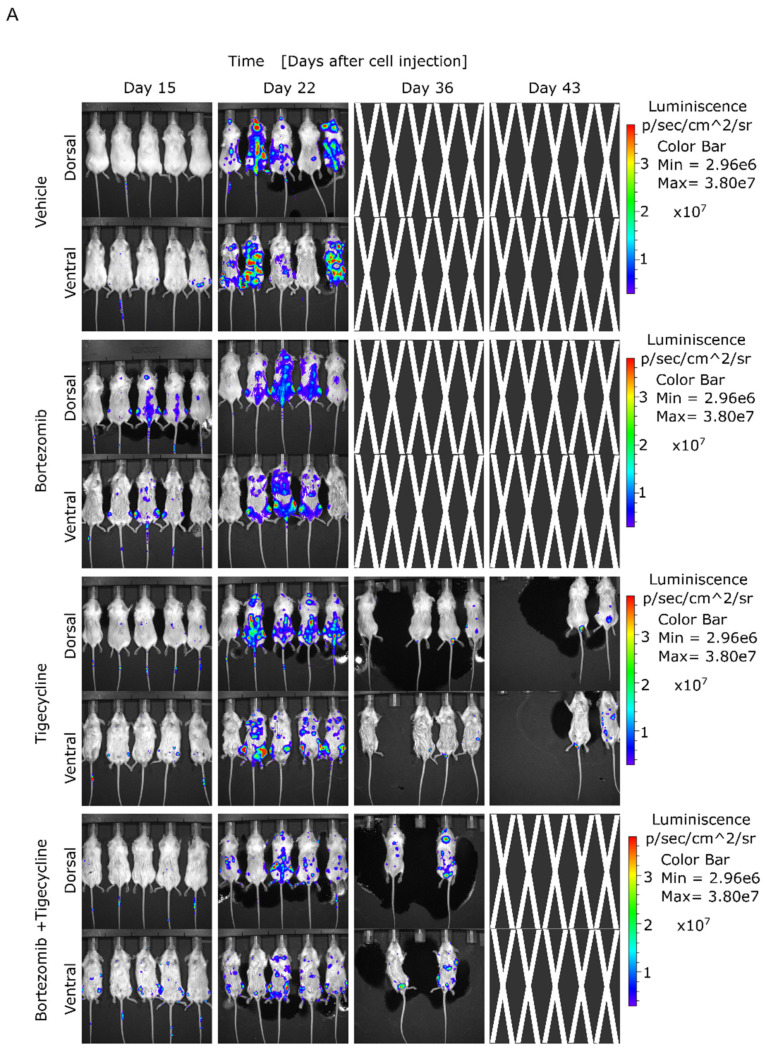
Tigecycline and the combination with bortezomib inhibits cancer cell growth in a MM-xenograft model. (**A**) Bioluminescence images were acquired using an in vivo imaging system. (**B**) Kaplan–Meier survival analysis of mice after the various treatments. ** *p* < 0.001, *** *p* < 0.0001.

**Table 1 cancers-13-01662-t001:** Main characteristics of multiple myeloma (MM) patients included in the study (*N* = 83).

MM Patients Clinical Variables		MGUS (*N* = 15)	SMM (*N* = 19)	NDMM (*N* = 27) ^a^	RRMM (*N* = 22)
Median age (range), years		65 (48–82)	76 (54–85)	68 (50–86)	65.5 (38–79)
Sex, (%)	Male	40	36.8	59.3	50
	Female	60	63.2	40.7	50
PC BM, average % (range)		7.87 (3–17)	22 (1.7–55)	39.2 (1–84)	42.4 (4–90)
PPC MFC %, average (range)		1 (0.3–5)	5.3 (0.7–24)	9.2 (1.4–38.7)	15.4 (0.03–40)
Type of Ig heavy chain (serum)	nondetected	0	0	3.8	0
	IgG	46.7	61.1	69.2	68.2
	IgA	53.3	38.9	26.9	18.2
	IgM	0	0	0	13.6
	IgD	0	0	0	0
	biclonal	0	0	0	14.8
Type of Ig light chain (serum)	nondetected	0	0	0	0
	kappa	25	44.4	74.1	50
	lambda	75	55.6	25.9	45.5
	biclonal	0	0	0	4.5
Serum M-spike, ≥3 g/dL (%)		0	13.3	19	25
Urine M-spike, detected (%)		14.3	18.2	60	53.3
Kappa, (%)	≥19.4 mg/L	77.8	40	57.9	50
Lambda, (%)	≥26.3 mg/L	66.7	40	36.8	41.7
Free kappa/lambda ratio, (%)	<0.26 mg/L	22.2	13.3	21.1	30
	≥0.26 <1.65 mg/L	22.2	53.3	0	0
	≥1.65 mg/L	55.6	33.3	78.9	70
Creatinine (%)	≥1.3 mg/dL	33.3	21.1	30.8	13.6
Serum calcium (%)	≥11 mg/dL	0	5.3	4	0
LDH (%)	≥225 U/I	14.3	21.1	12	23.8
Albumin (%)	≤2.8 g/dL	0	5.3	15.4	4.5
Immunoparesis, yes (%)		18.2	35.7	84.2	87.5
Refractory, yes (%)		NA	NA	31.8	50
Type of prior treatment (%)	with PI	NA	0	0	59.1
	without PI	NA	100	100	40.9
Type of following treatment (%)	with PI	NA	31.6	44.4	68.4
	without PI	NA	68.4	55.6	31.6
Best response categories *, (%)	VGPR	0	5.2	3.7	0
	PR	0	15.8	14.8	40
	CR	0	5.3	25.9	25
	CR MRD+	0	0	14.8	5
	CR MRD−	0	0	11.1	5
	SD	0	0	3.7	25
	NA	100	73.7	25.9	0

Percentage frequencies of several clinical parameters from sample patients used in this study. Abbreviations: BM, bone marrow; CR, complete response; Ig, immunoglobulin; LDH, lactate dehydrogenated; MRD−, minimal residual disease negative; MRD+, minimal residual disease positive; NA, not applicable; NDMM, newly diagnosed multiple myeloma; PC, plasma cell; PI, proteasome inhibitors; PR, partial response; RRMM, relapsed multiple myeloma; SD, stable disease; VGPR, very good partial response. * Objective response was assessed at the start of each cycle and confirmed by the physicians, using IMWG criteria. ^a^ NDMM patients englobe newly diagnosed and follow-up patients.

**Table 2 cancers-13-01662-t002:** Classification of samples included in the study (*N* = 110) according to the analysis performed.

Method	MGUS (*N* = 15)	SMM (*N* = 19)	NDMM (*N* = 27)	RRMM (*N* = 22)
Total samples, N	21	21	42	26
gene expression	9	9	18	11
IHC	4	7	13	11
HE	4	5	5	4
MFC	4	0	6	0

Abbreviations: HE, histoenzymatic; IHC, immunohistochemistry; MFC, multiparametric flow cytometry.

## Data Availability

The RNA-Seq data from the CoMMpass that were analyzed in this study are publicly available through the Multiple Myeloma Genomics Initiative (https://research.themmrf.org, accessed on 22 April 2020) with the identifier IA14.

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
