# Peer review of "Myc-Related Mitochondrial Activity as a Novel Target for Multiple Myeloma"

_cancers, 2021, doi:10.3390/cancers13071662_

Round 1

Reviewer 1 Report

The manuscripts “Myc-related mitochondrial activity as a novel target for multiple myeloma” by Ortiz-Ruiz et al. deals with the important concept of mitochondrial biology in multiple myeloma. The paper includes valuable studies on primary myeloma cells. However, there are several inconsistencies with the published literature and even within the manuscript.

Major concerns:

  1. Myc increases mitochondrial activity, but whether this particular phenomenon represents a weakness that can be exploited remains unclear. For the authors to make the claim that high Myc activity increases sensitivity to a drug, they need to show a titration of Myc levels within a given cell line and that their drug of choice preferentially targets Myc-high cells. At present, the connection between Myc levels and multiple myeloma cells’ susceptibility to tigecycline remains tenuous.
  2. My biggest concern is the mode of action of tigecycline. As an inhibitor of prokaryotic ribosomes, it blocks translation in mitochondria. The half-life of mitochondrial proteins is in the hour range. How could a drug of this type show an effect within minutes (Fig. 3F – I have the same concern for the activity of proteasome inhibitors in this assay)? Could tigecycline activate mitochondrial apoptosis instead? Do other mitochondrial drugs synergize with proteasome inhibitors (e.g. oligomycin, metformin)? Importantly, if the antibiotic shows such a dramatic effect, how come nobody noticed before that multiple myeloma patients treated for infection with a tetracycline derivative are benefiting and show reduced cancer growth?
  3. The discussion contains several inconsistencies. For instance, I am not sure proteasome inhibitors really show an increase in respiration (Fig. 3F seems to show a reduced level with BTZ compared to control treatment, even though the use of this drug is unlikely to affect respiration within the timeframe the authors chose). Most labs studying this topic have likely observed reduced respiration upon acute proteasome inhibition (6-48 hours). The increased respiration in proteasome inhibitor-resistant myeloma cells that the authors cite is an effort by the cell to bypass and compensate for the effects of the drugs, not a direct effect of the drugs. Also, while MYC is a short-lived protein that is acutely stabilized by proteasome inhibitors, the authors clearly show that the levels are diminished (Fig. 3C). This should be discussed in the paper. Likewise, they can not claim that MYC levels are elevated to argue proteasome inhibitors increase mitochondrial activity as it conflicts with their own data as shown in Fig. 3C (line 469-471).

Minor concerns:

  1. A short introduction on tigecycline would be helpful.
  2. The study on CD38/Daratumumab is lacking a rationale. Why did the authors pursue this in the context of their research?
  3. Figures: Some p values are missing. In general, figures should look like 1B, even though here the legend on the use of asterisks is lacking.
  4. 1A would be more convincing if the authors showed a directionality of the MYC effect (e.g. when comparing the top vs bottom 25% and 10%, the effect should be even stronger than comparing top vs bottom median).
  5. Fig. 4 requires more detailed legends. Is Fig. 4A showing a representative FACS experiment and 4B (left) the overall comparison? How many cells were compared? How long were they treated? Were primary myeloma cells grown in regular RPMI/FCS for 2 days or did they receive additional cytokines?
  6. Fig. 5: why did BTZ show no effect?

Reviewer 2 Report

In the present study, Ortiz-Ruiz A. et al. have investigated the impact of mitochondrial activity in MM development and tested the therapeutic efficacy of an OXHPOS inhibitor, tigecycline, to overcome proliferation of this MYC-driven tumor. In line with previous data, COXII expression, COX activity, mitochondrial mass and mitochondrial membrane potential resulted increased during disease progression thus suggesting progressive increase of mitochondrial features during tumorigenesis. As result, in vitro and in vivo use of tigecycline showed marked anti-MM activity alone and in combination with the frontline anti-MM drug bortezomib. Overall, Authors demonstrate a deregulated mitochondrial activity during MM transformation and progression which in turn results in striking benefit of therapeutic strategies targeting this vulnerability. Overall study identifies a novel weakness of Myc through mitochondria with meaningful therapeutic implications.

This is a well conducted study with quite interesting results, which should be more addressed to fully support Author’s conclusion

These are my comments:

  • A OXHPOS levels screening analysis should be performed in MM cell lines and ex vivo cells. Indeed, this analysis will help to better support oxidative capacity of MM cells
  • It is quite difficult to follow the role of CD38. Could you please describe the role of this ectoenzyme in affecting mitochondrial activity of MM cells? Otherwise please remove this part from current study.
  • Authors perform mitochondrial mass and membrane potential analysis of MM cells during progression and in premalignant stage MGUS. It could be more informative analyzing these features at diagnosis and during disease progression at RRMM stage to better address mitochondria role during tumor progression. In my opinion the use of pre-malignant stage is questionable for this purpose.
  • Authors screen a panel of HMCLs to determine the most appropriate model for mitochondrial inhibitors testing. Across screened cells emerges U266 cell line which is previously reported to expresses MYCL (L-MYC) but not either genomic rearrangement affecting the MYC locus nor MYC overexpression. Therefore, based on highlighted role fo MYC on mitochondrial activity deregulation, it could be intriguingly to employ genetically engineered cells. Otherwise U266 cells should be excluded from the screen
  • Figure 3 A shows tigecycline activity on MM cells alone and in combination with bortezomib. Unfortunately, single agent bortezomib data are missed in figure.
  • In the Figure 3 C authors indicate a slight increase in OXPHOS protein triggered by bortezomib treatment. What protein authors refer to? Also, COX activity showed in next panel does not differ in co-treatment compared with tigecicline therapy alone; please specify.
  • Figure 3E is quite difficult to follow. Could you please include MFI analysis for each stimulus?
  • Remarkably authors analyze the mitochondrial inhibitory effect of tigecycline on bortezomib resistant cells. It would be helpful supporting these findings by showing analysis of mitochondrial features (COXII expression, COX activity, mitochondrial mass, and mitochondrial membrane potential) in Bz resistant cells in parallel with isogenic cells
  • To confirm that mitochondrial activity blockade is an effect of a c-Myc driven mechanism inhibition, authors examine effects of a bromodomain BRD4 inhibitor. In such a scenario, as stated above, the U266 cell lines represents an alternative model for this analysis
  • Again, the role of CD38 in the entire story remains not fully explored and discussed otherwise it would be better to remove this data from present study to make it more focused on
  • Authors tested the efficacy of tigecycline alone or in combination with bortezomib on primary BMPCs collected from patients. Figure 4A shows activity of each stimuli on CD38+/CD138+ cells. It is opinion of this referee that the anti-MM activity of tested drugs should be supported by Annexin V /PI staining on the selected cells population.
  • In vivo data are misleading. It is quite interesting to see that bortezomib treatment alone does not affect mice survival but results in poorer outcome compared with vehicle control. Similarly, best responses in term of OS are obtained with tigecicline alone and not in its combination with bortezomib. These data are in contrast with in vitro data as well as with tumor growth experiments showed in panel A. Please correct these data to make in vivo results more consistent.
  • A reduced mitochondrial respiration in venetoclax-sensitive MM cells has been recently identified, with low electron transport chain (ETC) Complex I and Complex II activities associated with drug sensitivity. Based on these data ETC inhibition worth to be tested also in this context (in combination with bortezomib) to make study more updated.

Minor points

  • In Figures as well as throughout test Its better use term NDMM to refer diagnosis and RRMM for indicate relapsed/refractory stage.
  • Figure 1 C is quite difficult to understand. Positive cells are indicated as grade I or II. What grade I/II is for? Please specify
  • Please check typos mistakes and text spacing throughout the manuscript.
  • Standard deviation is missing in several experiments

Round 2

Reviewer 1 Report

Thank you for addressing all concerns. The manuscript now has tight focus and I do not feel a need to repeat the Seahorse assay after 48 hr or add the figures that the authors kindly shared in their rebuttal letter.